# Enabling Bitwise Reproducibility for the Unstructured Computational Motif [†]

**Bálint Siklósi** [1,*,‡,§] , **Gihan R. Mudalige** [2] and **István Z. Reguly** [1,‡,§]

1 Faculty of Information Technology and Bionics, Pázmány Péter Catholic University, 1088 Budapest, Hungary; reguly.istvan.zoltan@itk.ppke.hu
2 Department of Computer Science, University of Warwick, Coventry CV4 7AL, UK; g.mudalige@warwick.ac.uk
* Correspondence: siklosi.balint@itk.ppke.hu
† This paper is an extended version of our paper published in Proceedings of the 2020 20th IEEE/ACM International Symposium on Cluster, Cloud and Internet Computing (CCGRID), Melbourne, VIC, Australia, 11–14 May 2020.
‡ Current address: Práter street 50/a, 1083 Budapest, Hungary.
§ These authors contributed equally to this work.

**Abstract:** In this paper we identify the causes of numerical non-reproducibility in the unstructured mesh computational motif, a class of algorithms commonly used for the solution of PDEs. We introduce a number of parallel and distributed algorithms to address nondeterminism in the order of floating-point computations, in particular, a new graph coloring scheme that produces identical coloring results regardless of how many parts the graph is partitioned to. We implement these in the OP2 domain specific language (DSL) and show how it can be automatically deployed to any application that uses OP2 without user intervention. We contrast differences in results without reproducibility and then demonstrate how bitwise reproducibility can be gained using our methods on a variety of applications including a production CFD application used at Rolls-Royce. We evaluate the performance and overheads of enforcing bitwise reproducibility on a cluster of CPUs and GPUs.

**Keywords:** floating-point; bitwise reproducibility; unstructured-mesh computation; DSL; CPU; GPU; MPI

## 1. Introduction

Floating-point number representation and calculations form the backbone of science and engineering computations. They allow one to represent and approximate the continuous ranges of quantities/values on discrete systems such as digital computers. However, any floating-point representation, including the IEEE floating-point standard, by its very nature has finite precision and suffers from truncation errors, which makes operations on them non-associative [1]. This is particularly obvious when representing numbers that fall in large dynamical ranges. For example, the expression $(a + b) + c$ has a different answer than $a + (b + c)$ when, for example, $a = 10^{20}$, $b = -10^{20}$ and $c = 1$ with a 64 bit representation; $(10^{20} + -10^{20}) + 1 = 0 + 1 = 1$ in the former case and $10^{20} + (-10^{20} + 1) = 10^{20} + -10^{20} = 0$ in the latter. The issue is compounded in parallel systems, where the associativity is applied in a non-deterministic manner. Thus, the exact truncation events and the order in which they are performed lead to slightly different results. Over long executions, the errors accumulate, potentially leading to larger inconsistencies between results from multiple runs.

Execution of parallel applications and the results produced by their floating-point number computations lead us to the notion of numerical reproducibility. In the strictest sense, this means obtaining bitwise identical results from multiple runs of the same code that consume the same inputs. A less stringent requirement would be to accept results with errors less than machine precision, leading to the need for a tolerance range for

results. However, strict bitwise reproducibility could be essential for some applications, with many codes [2–7] implementing algorithms and techniques to enforce the required accuracy. Bitwise reproducibility is also useful for validating ported codes between different architectures. For example, by turning off fused multiply-add (FMA) operations and other optimizations, we can compare the output of a new GPU implementation to a previously trusted (validated) CPU implementation. If both produce the same result, then there is a high probability that we have managed to create the new version without introducing new errors. For relative debugging we already have examples of automatic test environments [8], with bitwise reproducibility, we can avoid the problem of choosing the margin of error.

Numerical reproducibility stands as a vital concern in the landscape of parallel computing, where the attainment of bitwise identical results across multiple executions is a sought-after goal. This emphasis on reproducibility becomes especially significant in computational domains like computational fluid dynamics (CFD), where the reliability and accuracy of conclusions drawn from simulation results are of paramount importance. In practice, the application of various algorithms and techniques in codes addressing challenges such as the wind vulnerability of structures [9] and modeling nonlinear aeroelastic forces [10] underscores the broader need for ensuring that the conclusions derived from these simulations are not only insightful but also reproducible.

Bitwise reproducibility, however, often comes at a performance cost. Time spent on carrying out order-preserving techniques to obtain identical results adds additional overhead, compounding total time-to-solution. As such, careful trade-offs should be considered depending on the application domain and the validation and performance requirements. Much of the current literature focuses on providing one-off solutions to this problem for specific applications. Many of them rely on Kahan's compensated solution method [11], where after adding up the high-order parts of two elements, the low-order error is stored and is accumulated with the low-order part of the next summation. Many apply this method specifically in their own applications [3,4,7]. Another widely used method is introduced by Demmel et al. [12] where a variable number of bins are created for different magnitudes and then are used to accumulate the given magnitude part of the operands. We can see some examples of the usage of Demmel's method in [3,13]. Other application-specific methods also exist, for example, sorted particle potentials in [6] and the use of integer conversions as in [14]. Most of these solutions require altering the code manually, often using a different number representation, making code maintenance difficult and expensive. This is especially problematic for large codebases. Additionally, most of these solutions address a single target architecture, making it even more laborious and costly when aiming to develop and maintain a performance-portable application.

The underlying goal of this paper is to explore the challenges in achieving reproducibility, specifically bitwise reproducibility for the domain of unstructured mesh computations, one of the seven dwarfs [15] in HPC. The distinctive feature of unstructured mesh computations is the existence of data-driven indirections (such as mapping from edges to vertices) and computations that indirectly increment/read-write data, which causes data to race in a parallel environment. Although we are not aware of a systematic approach for unstructured meshes, we can see a number of similar works for other domains. The reproBLAS project [16] covers many use cases in the field of dense linear algebra with reproducible execution. Apostal et al. [17] created a code scanner, which can automatically recognize certain reductions where reproducibility might cause problems. In this paper, on top of providing a general solution applicable to a wide range of unstructured-mesh applications, we showcase how reproducibility can be implemented within the OP2 domain specific language (DSL). This paper is an extension of early work demonstrating the temporary array method on two simple benchmarks [18]. Our results enable us to deliver reproducibility automatically to a number of existing applications, written using OP2, including a full-scale industrial computational fluid dynamics (CFD) code, executing on both CPU and GPU cluster systems. Specifically, we make the following contributions:

1.  We identify key sources of non-determinism in unstructured mesh computations and propose three techniques for addressing them for this domain: (1) use of temporary arrays for indirect increments, (2) coloring for indirect increments and read-writes, and (3) reproducible reductions.
2.  To use a coloring approach for reproducible execution, we develop a deterministic coloring algorithm, which depends only on the mesh and is independent of the partitioning of the mesh (including the number of partitions).
3.  The above-developed techniques and algorithms are implemented within the OP2 DSL, in order to automatically generate target-specific parallel code that produces reproducible results when executed on modern large-scale systems with multi-core and many-core processor architectures. Leveraging OP2's source-to-source translation, we can deliver bitwise reproducibility without changes to the user code.
4.  Various unstructured mesh applications, ranging from smaller benchmarks (Airfoil [19], Aero [20]), a CFD mini-app (MG-CFD [21]) to a large-scale industrial CFD application (Rolls-Royce Hydra [22]), previously developed with the OP2 DSL are used to evaluate our proposed algorithms. Numerical results as well as the impact on performance when executed on CPUs, GPUs, and their scalability on clusters are explored.

To the best of our knowledge, our work is the first to provide a general solution for bitwise reproducibility on unstructured mesh applications. We show that this solution achieves good results in terms of accuracy and performance in industrial applications, such as Rolls-Royce Hydra, demonstrating the practicability of this work for production codes.

The rest of this paper is organized as follows: in Section 2 we discuss related works, in Section 3 we present background on floating-point number presentations and computations and introduce the sources of non-reproducibility, with examples from a number of applications. In this section, we also describe the unstructured mesh application class and OP2's abstraction and framework. In Section 4 we describe multiple methods with which we achieved bitwise reproducibility. In Section 5 we examine the performance of these techniques, and in Section 6 we draw conclusions. The codes developed for this paper are available in the Supplementary Materials, which were accessed on 3 January 2024.

## 2. Related Works

Bitwise reproducibility is a widely researched problem, usually investigated in a specific application.

Mascagni et al. [2] list the main sources of non-reproducibility in a neuroscience application: (i) the introduction of floating-point errors in an inner product; (ii) the introduction of floating-point errors at each an increasing number of time steps during temporal refinement (ii) and (iii) differences in the output of library mathematical functions at the level of round-off error. They highlight the importance of numerical reproducibility without providing a general solution.

Liyang et al. created a special method [6] for molecular dynamics applications in the LAMPPS Molecular Dynamics Simulator [23]. From each particle, the potentials are calculated first and then stored temporarily. Then they loop over every particle again, sort the components for one element, and accumulate them in ascending order. This way, they were able to eliminate the effect of non-associative accumulation.

Langlois et al. [3] tested multiple techniques for reproducible execution on an industrial free-surface flow application: the 2D simulation of the Malpasset dam break. All methods passed, but their main purpose is to determine how easy it is to use them. Kahan's compensated solution method [11] appeared to be the easiest to apply and provided accurate results for low computing overhead. The integer conversion provided in Tomawac [14] was also easy to derive and introduced a low overhead. The solution that uses reproducible sums [12] was efficient, but was applied less easily in their case and introduced a significant communication overhead.

He et al. [4] experimented on a dynamical weather science application. They tested several methods, such as Kahan's [11], or the double-double number technique [24] which

is an unevaluated sum of two IEEE double precision numbers. They also provide an MPI operator for reductions.

Taufer et al. [5] were looking into a molecular dynamics application, whereby reproducibility meant that results of the same simulation running on GPU and CPU lead to the same scientific conclusions; in their case, bitwise reproducibility was not necessary. They tried double precision arithmetic, which partially corrected the drifting, but was significantly slower than single precision, comparable to CPU performance. They created a library of float-float composite type, which is comparable in accuracy to double, but the performance loss is only 7%, versus a loss of 182% of normal double precision.

Robey et al. also experimented with a dynamical fluid application [7]. They tried to sort their data first and then sum, but that was too slow. They applied Ozawa's pair-wise summation [25], which produced less truncation, but not bitwise reproducibility, although this method is quick and can run in parallel. The double-double technique used too much memory, so finally they used Kahan's [11] and Knuts's [26] approach due to their simplicity, low additional cost and their added precision.

Apostal et al. [17] developed a source code scanner to recognize reductions over MPI in C or C++ codes and automatically modify them to use Kahan's summation [11] or an algorithm developed by Demmel and Nguyen [12].

Olsson et al. [27] defined some transformation techniques to describe concurrent applications written in the SR programming language to achieve reproducibility. They can transform an arbitrary SR program into two parts: one for recording a sequence of events and one for replaying those events.

Reproducible Basic Linear Algebra Subprograms [16] (ReproBLAS), intends to provide users with a set of parallel and sequential linear algebra routines that guarantee bitwise reproducibility independent of the number of processors, data partitioning, reduction scheduling, or the sequence in which the sums are computed in general. The BLAS are commonly used in scientific programs, and the reproducible versions provided in the ReproBLAS will provide high performance while reducing user effort for debugging, correctness checking, and understanding the reliability of programs.

Graph coloring is a widely used method in HPC to maximize parallel efficiency, without facing any race conditions. We can see a detailed example of using coloring techniques in the work of Zhang et al. [28]. Their paper addresses challenges in parallelizing unstructured CFD on GPUs, employing graph coloring for data locality optimization and parallelization, resulting in substantial speed-up with GPU codes outperforming serial CPU versions by 127 times and parallel CPU versions by more than thirty times in the same MPI ranks.

## 3. Background

### 3.1. Floating-Point Representation

The IEEE-754 [29] standard specifies the format for representing floating-point numbers, as well as rounding modes and arithmetic operations such as addition $\oplus$, subtraction $\ominus$, multiplication $\otimes$, division $\oslash$ and square root *sqrt*. Floating-point numbers are written as $x \cdot 2^E$, where the mantissa $x \in [1, 2)$ is a number of $m$ binary digits (bits) and $E$, $E_{min} \leq E \leq E_{max}$, is an integer called the exponent. The format specifies $m$, $E_{min}$ and $E_{max}$. Representable numbers are those that can be expressed in this notation. If the solution of an operation is not representable with one setup, then the result is rounded to a representable number. The round-to-nearest, round-towards-zero, round-towards-positive-infinity, and round-towards-negative-infinity rounding modes are selectable by modifying the internal state of the floating-point unit (FPU). Every rounding mode has a rounding function $fl(x)$ that converts a real number $x$ into a representable number. Every arithmetic operation is defined as the rounding of an abstract arithmetic operation's exact outcome. For example, $fl(a + b)$ will become the result of the sum of two integers $a$ and $b$. All intermediate values are rounded in computations that involve more than one operation. As a result, the operators $\oplus$ and $\ominus$ are not associative. When $a = 10^{20}$, $b = -10^{20}$, and $c = 1$ with a 64-bit representation, the expression $(a + b) + c$ yields 1 while $a + (b + c)$ produces 0. We have to

note here that the differently accumulated roundoff error in most cases should not change the validity of an application [30]. Changing the execution order of an algorithm may still produce valid results, independently of being reproducible.

We can observe this effect in Figure 1, using a more realistic finite element method example with a conjugate-gradient solver, with calculations in double precision (Aero [20]—detailed in Section 3.8). On this histogram, we counted the number of different values of the end results in relative differences for several magnitudes between running the application using eight processes and 16 processes. From the 6.5M elements, there were only 3599, which had a bitwise identical result, the rest had a difference between $10^{-12}$ and $10^{-4}$, and most of them were around the magnitude of $10^{-8}$.

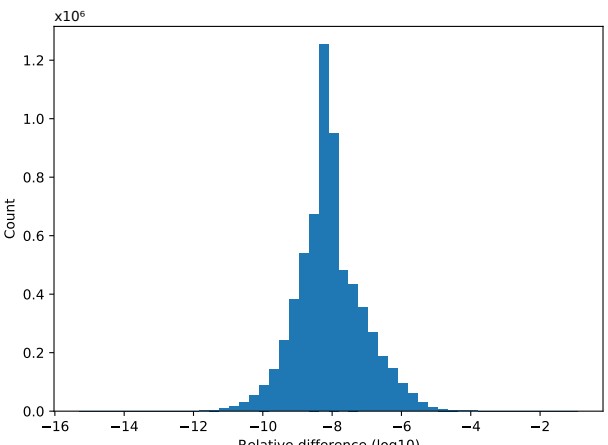

**Figure 1.** Histogram, showing the relative differences in a conjugate-gradient solver (Aero) between runs with eight processes and 16. The result converges to a numerically stable state, but on average there is a $4.05 \times 10^{-7}$ difference.

### 3.2. Reproducibility

Reproducibility is often understood as experimental reproducibility. This is also a widely researched topic [31–35], but our aim is to obtain bitwise identical results of an application run with the same input parameters regardless of the level of parallelism, be it the number of threads or processes executed simultaneously. Non-reproducibility is not caused by the roundoff error but by the non-determinism of accumulative roundoff error. Due to the non-associativity of floating-point addition, accumulative roundoff errors depend on the order of evaluation, which is almost always relaxed in parallel and distributed environments. In a distributed MPI environment, there are multiple possible sources of non-associativity: number of MPI nodes, MPI reduction tree shape, number of cores per node, and data ordering. The histogram in Figure 1, which runs the Aero benchmark of the OP2 library, shows the relative differences ($\frac{(a-b)}{a} \mid a > b$) of a non-reproducible application run with different numbers of MPI processes. In general, some of the causes might be efficiently addressed, such as the reduction tree shape, which can be defined by network interface cards [36], but changing the number of processes can cause issues that are not as easily addressed. A general solution might be to fix the order of evaluation but that is, in many cases, incompatible with parallelization, and running sequentially is prohibitively costly. Another solution is to eliminate rounding errors. We can use exact arithmetics [37], but that will substantially increase the memory usage and the cost of the computations, as well as the amount of communication when applied to more complicated operations such as matrix multiplication. Higher precision can be used, but it will be reproducible only with higher probability [38].

### 3.3. Reproducible Reductions

One of the most common sources of non-reproducibility comes from reductions, where we add up the elements of an array into a single result. When carrying this out with a

parallel execution, the rounding errors can accumulate rapidly. There are multiple solutions for this problem [12,16,39], but the underlying observation is common to all approaches; adding up numbers with similar magnitudes is going to be exact. Demmel et al. [12] use pre-roundings to a well-calculated magnitude with an extra sweep through the array, add the values together, and then apply the same method on the remainders of the roundings. Arteaga et al. [39] extended their work by calculating the magnitudes without the additional sweep. The ReproBLAS library [16] creates bins for the magnitudes in advance and uses them in parallel for the summations. In our project, we use ReproBLAS, due to its user-friendly implementations; though the necessary reductions can be calculated by using other techniques as well.

### 3.4. ReproBLAS

Reproducible Basic Linear Algebra Subprograms [16] (ReproBLAS), intends to provide users with a set of parallel and sequential linear algebra routines that guarantee bitwise reproducibility independent of the number of processors, data partitioning, reduction scheduling, or the sequence in which the sums are computed in general. It assumes that floating-point values are binary and conform to IEEE Floating-Point Standard 754-2008, floating-point operations are conducted in ROUND-TO-NEAREST mode (ties may be broken at will) and that underflow happens gradually. Summing $n$ floating-point values with their default settings costs around $9n$ floating-point operations (arithmetic, comparison and absolute value). The new "augmented addition" and "maximum magnitude" instructions in their proposed IEEE Floating-Point Standard 754-2019 [40] can theoretically reduce this count to $5n$. On a single Intel Sandy Bridge core, for example, the ReproBLAS slowdown compared to a performance-optimized non-reproducible dot product is $4\times$ [41]. Here, the output is reproducible regardless of how the input vector is permuted. For the summing of 1,000,000 double-precision floating-point (FP64) values, the slowdown on a large-scale system with more than 512 Intel "Ivy Bridge" CPUs (the Edison machine at NERSC) is less than $1.2\times$. The result is also reproducible regardless of how the input vector is partitioned across nodes or how the local input vector is stored within a node.

### 3.5. The Unstructured Mesh Computational Motif

Computations defined on unstructured meshes form an important basis for many engineering calculations commonly used in PDE discretizations, such as finite elements of finite volumes. An unstructured mesh is characterized by a number of sets (vertices, edges, cells, etc.) with explicit connectivity information between them (e.g., edges to vertices). Computations are commonly expressed as a parallel loop over a set, with computations accessing data either directly on the iteration set or through an indirection. For example, a common operation in computational fluid dynamics is to compute fluxes across faces (edges), and then increment/decrement state variables defined on connected cells. The key motif here is the edge-centered computations indirectly incrementing cell data, which then gives rise to non-determinism when the order of execution of the edges is relaxed for the sake of parallelism. Another common pattern is the global reduction, often conducted in a non-deterministic order, where the result is then used in subsequent computations. For example, in the conjugate gradient algorithm, the results of dot products are used as weights in the next step.

The distributed and parallel execution of unstructured mesh algorithms is a well-established field [42–45]. For distributed memory execution, the mesh is partitioned using one of many established libraries, such as PT-Scotch or ParMetis [46,47]. It is important to note here that an unstructured mesh is a hypergraph, consisting of multiple "vertex" types, whereas most partitioners only partition a simple graph, and the rest of the hypergraph is usually partitioned in a greedy way through connections to the simple graph. This is then related to how computations are executed, an "owner-compute" approach is commonly utilized, where all computations associated with a given element are performed on the process that owns that element. So, for instance, in the earlier example, the process that

owns a given cell will execute all the edges that increment that cell, even if some of those edges are not owned by it. This requires communicating all the data needed to execute those edges as well. This often leads to redundant computations around partition boundaries. Depending on the exact implementation the deterministic order of execution for elements is often relaxed at this point to allow shared memory parallelization and powerful optimizations such as overlapping computations and communications.

To enable shared-memory parallel execution of unstructured mesh computations, one needs to address the issue of race conditions when indirectly incrementing/updating data. Virtually all execution schemes used in the literature rely on the associativity of these operations, for example, by using atomic updates, a staging of increments in an auxiliary array and their separate sum, or a coloring scheme [48,49]. We are not aware of related works that explicitly aim to maintain an ordering of operations whilst enabling shared memory parallel execution.

### 3.6. OP2

The OP2 library provides a programming abstraction for describing unstructured meshes and computations on them, relying on the access-execute paradigm to separate the description of computations from the actual parallel implementation. OP2 defines sets, mappings between sets, and data on sets. Computations are then described as parallel loops over a given set, accessing data either on the iteration set or through at most one level of indirection. The type of access is also explicitly declared (read, write, increment). Based on such a description of computations, OP2 can automatically parallelize computations in both distributed and shared memory systems, such as multi-core CPUs and GPUs [45]. Thanks to the separation of per-element computations from how the execution of elements is scheduled and how data are moved, OP2 can take full control of how the computation is carried out on a processor. In the work presented in this paper, we utilize this to apply a variety of approaches that allow for the deterministic ordering of indirect accesses (increments and updates), guaranteeing the bitwise reproducibility of the results.

### 3.7. Automatic Code Generation for Reproducible Execution

Altering an already existing nonreproducible code to be reproducible might be tedious and laborious. Fortunately, in some ways, this process can be automated.

OP2 has an already established workflow to generate platform-specific optimized applications [50], Figure 2 summarizes the main mechanisms. If an application is implemented using OP2's API, then a source-to-source translator can generate platform-specific application files, which later can be compiled and linked with the backend libraries of OP2. In our current work, we modified three stages of the workflow. We added API calls to the application description, so the user can choose which reproducible strategy should be applied. In order to use these strategies, the source-to-source translator had to be updated to generate such application files that use the reproducible backend libraries with MPI or CUDA.

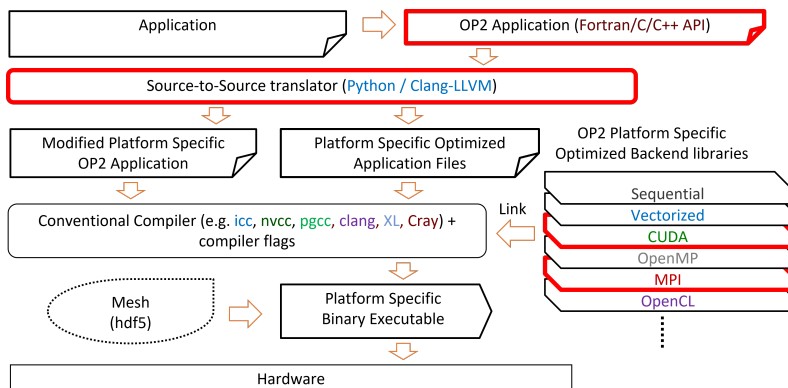

**Figure 2.** Flow diagram of the mechanism of OP2. The bold, red frames represent the updated steps of OP2's workflow from our work.

### 3.8. Test Applications

The following applications are implemented in OP2 to evaluate and assess the efficacy and performance of our proposed algorithms.

Airfoil [19] is a representative CFD code, written using OP2's C/C++ API. It is a non-linear 2D inviscid airfoil code that uses an unstructured grid. Airfoil uses a finite volume method to solve the steady-flow 2D Euler equations using scalar numerical dissipation. Airfoil is available as part of the OP2 framework.

Aero [20] is a 2D non-linear steady potential flow simulation of air moving around an airfoil, developed based on standard finite element methods. It uses a quadrilateral grid similar to that used by the Airfoil application but uses a Newton iteration to solve the non-linear equations defined by a finite element approximation. Each Newton iteration requires the solution of a linear system of equations. The assembly algorithm is based on quadrilateral elements and uses transformations from the reference square to calculate the derivatives of the first-order basis functions. Dirichlet-type boundary conditions are applied on the far-field, and the symmetric sparse linear system is solved with the standard conjugate-gradient (CG) algorithm. Aero is also available as part of the OP2 framework.

MG-CFD is a 3D unstructured multigrid, finite-volume computational fluid dynamics (CFD) mini-app for solving an inviscid flow problem. It performs a three-dimensional finite-volume discretization of the Euler equations for inviscid, compressible flow across an unstructured grid by extending the CFD solver in the Rodinia benchmark suite [51,52]. It accumulates fluxes by performing a sweep across edges, which is implemented as a loop over all edges. Multigrid support is achieved by supplementing the Euler solver's architecture in the work of Corrigan et al. [51] with crude operators that transport the simulation's state between multigrid levels. MG-CFD was originally created as a CPU-only implementation[53], but it has since been implemented with OP2 as well. It can be downloaded as open-source software [21].

Hydra [54] is a full-scale industrial CFD application for the design of turbomachine components of aircraft engines at Rolls-Royce. Hydra is a complex and configurable application that can perform various simulations on highly detailed unstructured meshes. Its development originally started 23 years ago [55], and it is still actively developed and optimized to this day. The simulations implemented in Hydra are typically applied to large meshes, which can contain tens to hundreds of millions of edges and can run from a few minutes to weeks. It consists of several components that simulate various aspects of the design, including the steady and unsteady flows that occur in the engine around adjacent rows of rotating and stationary blades, the operation of compressors, turbines and exhaust, and the simulation of behavior such as ingestion of ground vortices. The guiding equations to be solved are the Reynolds-Averaged Navier–Stokes (RANS) equations, which are second-order PDEs. By default, Hydra uses a 5-step Runge–Kutta method for the time-marching, which is accelerated by multigrid and block-Jacobi preconditioning [55,56]. Our work uses the Hydra setup with several configurations: an unsteady simulation of two blades of DLR's Rig250 mesh and a steady simulation of NASA's Rotor37 mesh with different turbulence models: the Spalart–Allmaras wall function model, which is a one-equation model that solves a modeled transport equation for the kinematic eddy turbulent viscosity and a k-$\omega$, which is a two-equation model that is used as an approximation for the Reynolds-averaged Navier–Stokes equations (RANS equations). Again, we highlight the effect of nonreproducibility on a few examples with Hydra. In Figure 3 we can observe how the relative difference accumulates when increasing the number of time-steps from 10 to 100 while using the same unsteady numerical method on the same mesh. For a full revolution of two blade rows, 2000 time-steps are needed, where one time-step contains 10 iterations. In Figure 4 we show that different turbulence models are impacted differently by the relaxation of the execution order, run for 100 iterations with a steady simulation on the NASA Rotor37 benchmark. The k-$\omega$ is more susceptible to rounding error than the Spalart–Allmaras. The variable $\omega$ is used to avoid singularity near the wall, but it also becomes more sensitive to precision than the Spalart variable. This has a knock-on

effect on the whole boundary layer, and hence the flow field. All four histograms present the magnitude of differences between two runs with the same setup, just running with different numbers of MPI processes.

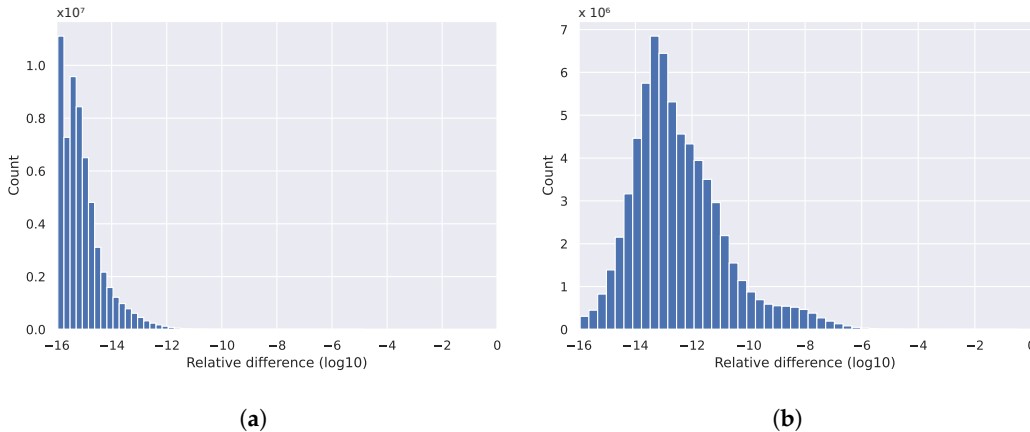

(**a**)                                                                                  (**b**)

**Figure 3.** Histograms, generated by using Hydra. The relative difference increases with more timesteps on an unsteady numerical solver. (**a**) Rig250 mesh with 20M nodes, 10 timesteps, Spalart–Allmaras model; (**b**) Rig250 mesh with 20M nodes, 100 timesteps, Spalart–Allmaras model.

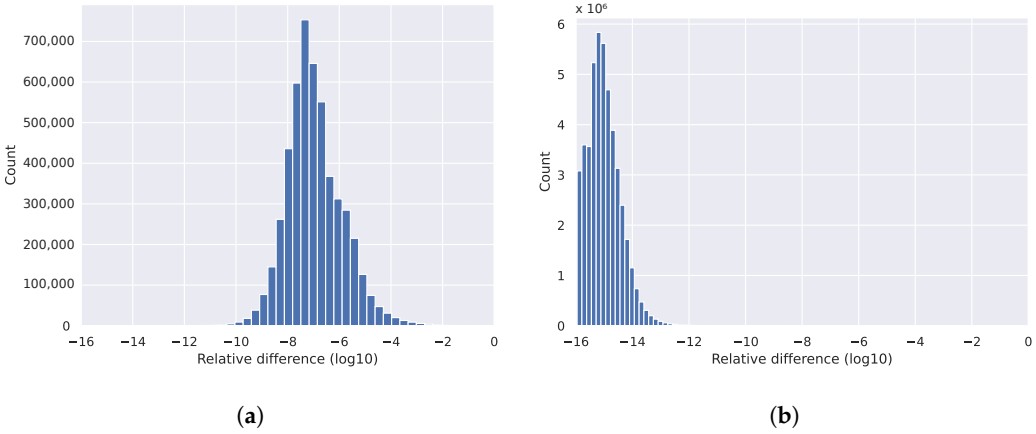

(**a**)                                                                                  (**b**)

**Figure 4.** Histograms, generated by using Hydra. The two models are not directly comparable, but they illustrate how the relative difference depends on the numerical properties of the applied model. (**a**) Rot37 mesh with 700k nodes, 100 iterations, k-$\omega$ model; (**b**) Rot37 mesh with 8M nodes, 100 iterations, Spalart–Allmaras model.

## 4. Theory and Calculation

In this section we describe our techniques to solve the two main problems which cause non-reproducibility, local element-wise reductions and global reductions. Most of our methods focus on the local reductions. For global reductions we utilize ReproBLAS. Most of our examples in this section use an edges→cells mapping, but all of these algorithms are implemented generally using the dimension of the specific mapping.

To solve the issue of ordering in local (element-wise) reductions, we provide two separate approaches: (1) a method storing increments temporarily and applying them later in a fixed order and (2) different reproducible coloring techniques, which later can be used as colored execution, maintaining deterministic ordering. For all of these techniques, we must provide a common deterministic seed that will always be the same, even with different numbers of MPI processes. That common seed is the global ID of all elements in the whole mesh. If there are multiple MPI processes, then the global IDs of each element must be communicated between the processes. If an element is owned by the given process, then its global ID can be looked up from an internal data array of OP2. If an element is

not owned, then its global ID must be imported from the MPI process that owns it. All of our techniques use two main parts: (1) the OP2 backend must calculate the execution order and (2) the generated code must execute the computations in this order. We apply the reproducible execution methods only on kernels where the order of summation does matter. These are loops with global reductions, indirect incrementing operations (OP_INC), or operations with an indirect read and write access pattern (OP_RW).

### 4.1. Temporary Array Method

A temporary array-based technique can be used to ensure reproducibility for incrementing operations. Consider using an edge→cells mapping and an incrementing operation. Here, we would iterate through all the edges, calculate values, and add them to a variable defined on a neighboring cell. To achieve reproducibility, we modify this structure by storing the calculated increments in a temporary array defined on the edges, and after all the increments are calculated, we iterate through all the cells and apply these increments in a fixed order defined by the global_IDs of the edges. In Figure 5 we can see an example of this method, where edge2's global_ID is the smallest, so the value from edge2 is applied first on the cell, then edge0, etc.

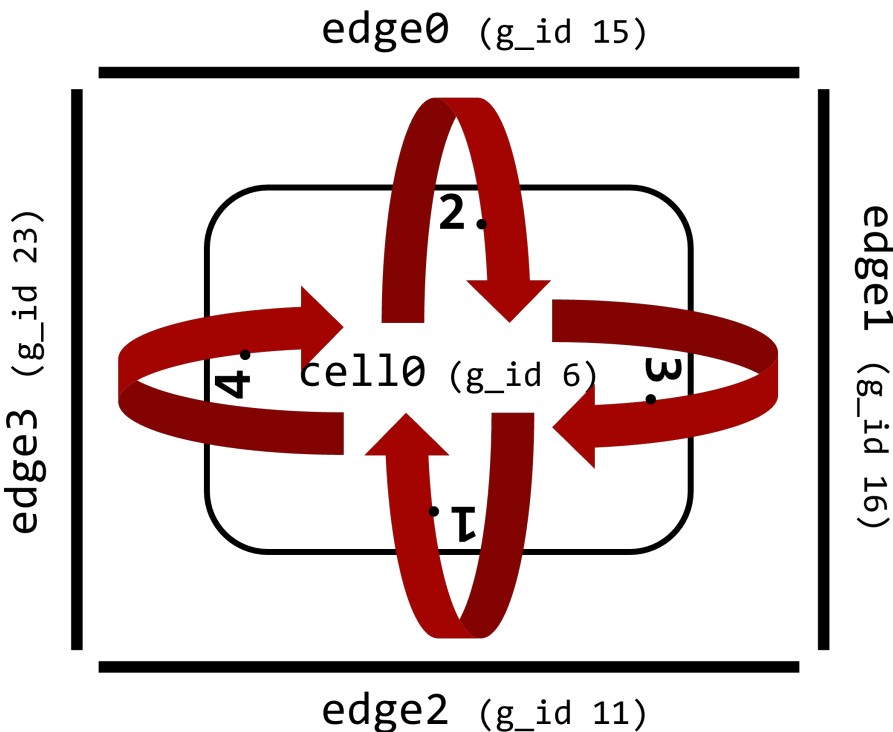

**Figure 5.** Example execution order of edges around a cell. Due to local id renumbering, the global ids must be used for a reproducible execution order.

To achieve this modified execution, a few extra preparations must be conducted in the backend, which are shown in Algorithm 1. After the global_IDs are shared, the next step is to create a reversed mapping for every map. The reversed mapping is needed so we can iterate through the cells and in each iteration we can access the edges connected to the given cell. This reversed map uses local indices which might be in different order in different MPI ranks. That is why we need to reorder them by using their previously shared global indices. Another modification conducted on the reversed map is that it actually stores indices of a temporary array where the increments from the edges are stored for a cell. In other words, if the *k*th element in line *n* (*k*th edge connection of cell *n*) of the

reversed map is *x*, then it means that in the temporary increments array at location *x* the increment for cell *n* from edge *k* can be found.

The main disadvantage of this method is the need for significant additional memory to store the reversed mapping, and to store the increments. The reversed map uses a Compressed sparse row (CSR) format, which consists of a main array of increment indexes (integers), with the size of `set_from_size` ∗ `original_map_dimension`, and another array indexing the previous array with a size of `set_to_size`+1. The temporary arrays themselves can use much more memory: `set_from_size` ∗ `map_dimension` ∗ `data_element_size`.

---

**Algorithm 1** Algorithm of generating incrementing order

---

exchange global IDs
*OP_map_index* = number of maps
**for** *m* = 0 to *OP_map_index* **do**
    create reversed mapping for map *m*
    *set_to_size* = target set's size of map *m*
    **for** *i* = 0 to *set_to_size* **do**
        sort the reversed connections of *i* by global IDs
    **end for**
**end for**

---

After creating the reversed map with the correct order, we generate a new `op_par_loop` implementation code to use this modified method. The main changes can be seen in Algorithm 2. After the initialization phase, it is imperative to set all elements in a temporary array to zero to accommodate individual increments. This step is crucial as the user kernel performs the increments, and proper initialization is required beforehand. Moreover, this approach ensures that the data remain in the cache, enhancing overall performance. Then we can call the kernel function for all edges to access the elements defined on the cells. If a parameter is accessed through an `OP_READ` or `OP_WRITE` method, then the execution order does not matter, so we can use the original method of directly storing the new state in the data. If the parameter is incremented (`OP_INC`), then we need to store each increment value in the `tmp_incs` array instead of adding it to the actual data. After the iteration on edges is completed and all increments are calculated, we need to apply those to the actual data on cells. For that, we start a new cell-based loop on the cells and by using the reversed mapping with the fixed ordering, for each cell, we can gather and apply the increments. This method is generally applicable to other types of mappings as well.

---

**Algorithm 2** Algorithm for applying the order of increments

---

*set_from_size* = source set's size of the original map
*original_map_dim* = the dimension of the original map
*set_to_size* = target set's size of the original map
**for** *n* = 0 to *set_from_size* ∗ *original_map_dim* **do**
    *tmp_incs*[*n*] ← 0
**end for**
**for** *n* = 0 to *set_from_size* **do**
    prepare regular access indices for *OP_READ* and *OP_WRITE* parameters
    call kernel function, using the *tmp_incs* array for *OP_INC* parameters
**end for**
**for** *n* = 0 to *set_to_size* **do**
    **for all** connection *i* of *n* **do**
        apply the temporary increment from connection *i* on the final location of the data
    **end for**
**end for**

---

### 4.2. Reproducible Coloring

The temporary arrays method only works for increment-type operations, where increments can be stored separately. If a kernel not only increments a variable but also reads and rewrites it (OP_RW), then the kernel call from one edge must be executed, storing its result in the cell before another edge accessing that cell can be executed. Although OP2 still requires that the computation be associative, we cannot store the increments separately. This problem needs a solution to be able to really execute the kernel calls in a predefined fixed order and achieve reproducibility. To solve this issue, we can apply a regular coloring scheme with the following restriction, we are looking for an equivalence class of colorings where if the color of one element is smaller than that of another connected element in the case of one coloring, then it should also be the same in the case of any other coloring.

We have three main approaches to solve this problem. An initial trivial solution is to choose the global index of the edge as the color. With this, we have as many colors as edges in any given subgraph, but we do not have multiple edges with the same color. This is useful for MPI-only parallelization, but not for a shared memory method. The advantage is that this trivial method can be solved without actually coloring the elements. We can just use the numbering from the global_ids for ordering sequential execution. This trivial execution schedule can be considered as a special case of colored execution and in fact they use the same generated code. Therefore, we refer to it as a coloring method. The second method is a non-distributed method, we apply a greedy coloring algorithm on the whole mesh in a single process as a pre-processing step and save the assigned colors in a file. When we rerun the application on multiple processes, we load and distribute the saved colors the same way as we distribute the mesh elements between the processes. With greedy coloring, we can generate a near-optimal number of colors, thus we have a high degree of parallelism. The drawback of this option is that we have to execute the pre-processing part in a single process. This carries the restriction that the whole mesh must be able to fit into the memory on a single node. The third method is a novel distributed coloring scheme, which does not suffer from this restriction.

Distributed Reproducible Coloring Method

We base our method on an algorithm developed by Osama et al. [57]. This original non-reproducible parallel method can be seen in Algorithm 3 between lines 7 and 40. We iterate through each element, calculate a local hash value and then compare it to its (as yet uncolored) neighbors' hash values. If the examined hash value is a local minimum or maximum in its neighborhood in a given iteration, then we can assign it a color. In our implementation we use Robert Jenkins' 32 bit integer hash function [58]. This hash function is a custom, non-cryptographic function that operates on unsigned integers. It uses a combination of bitwise operations and arithmetic with specific constants to compute the hash of an input.

The difficulty of applying this algorithm in a distributed graph comes from two sources. First, in each iteration of the previously described algorithm, we must know if the neighbor element already received a color, or not. Thus, we need to synchronize the assigned colored values on the borders of each subgraph (MPI partition). Secondly, it is difficult to figure out all the neighbors of an element on the border of a subgraph in a standard owner-computed model. We can see an example of this problem in Figure 6. Solid dots and continuous lines are the owned elements. In this example, we use an edge → nodes mapping, thus we import one layer of halo elements (e.g., edge 7, 8, 9 on Process 0) so we can update the owned nodes from all attached edges (so far it is a standard owner compute model). However, to calculate the smallest hash value in a neighborhood, we also need to communicate edges even around the non-owned nodes (e.g., edge 0, 2, 5, 6 on Process 1). Our extension to distributed execution can also be applied to other iterative coloring techniques that use only local information (the algorithm is not sequential) and are deterministic even with different graph partitioning. The number of colors is not explicitly minimized.

---

**Algorithm 3** Algorithm for reproducible coloring in a distributed graph

---

```
 1: create neighbor lists
 2: global_done = 0
 3: local_done = false
 4: if set_size == 0 then
 5:    local_done = true
 6: end if
 7: iteration = 0
 8: low_color = 0
 9: high_color = 1
10: while global_done < number of subgraphs do
11:    if not local_done then
12:       for all element e in from_set do
13:          if e has no color then
14:             calculate hash value of e in iteration i
15:             is_min = true
16:             is_max = true
17:             for all neighbors n of e do
18:                if n has no color then
19:                   calculate hash value of n in iteration i
20:                   if n's hash < = e's hash then
21:                      is_min = false
22:                   else if n's hash > = e's hash then
23:                      is_max = false
24:                   end if
25:                end if
26:             end for
27:             if is_min then
28:                give low_color as color of e
29:                number of noncolored elements − = 1
30:             end if
31:             if is_max then
32:                give high_color as color of e
33:                number of noncolored elements − = 1
34:             end if
35:          end if
36:       end for
37:       if number of noncolored elements == 0 then
38:          local_done = true
39:       end if
40:    end if
41:    exchange halo color values
42:    reduce local_done values into global_done
43:    low_color += 2
44:    high_color += 2
45:    iteration += 1
46: end while
```

---

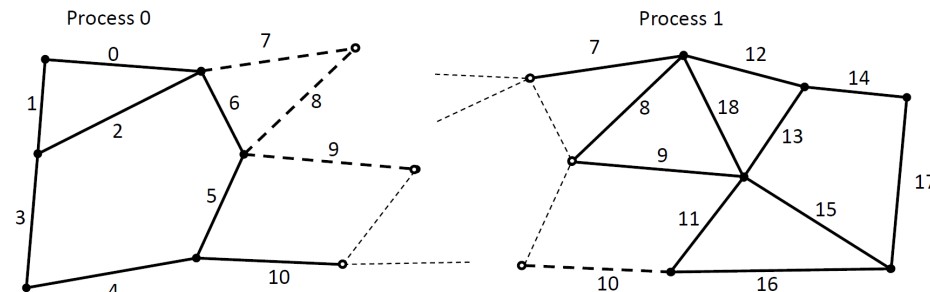

**Figure 6.** An example of a second ghost layer to determine the edge→edge neighbors on the partition borders. The numbers on the edges indicate their unique ID.

### 4.3. Parallel Global Reduction

Global reductions are another source of non-reproducibility in MPI applications. This operation is commonly conducted by performing a local sum on each process, then calling `MPI_Reduce`, however, this assumes associativity. If we use different numbers of MPI processes, then we would sum different elements and even a different number of elements locally, which again can produce different results. To solve this issue, we introduced another temporary storage. If a kernel performs an increment reduction, then we give a temporary storage point to store the increment for the result of each element. Then, in each MPI process, we reduce these increments reproducibly by using the ReproBLAS library. First, we create a local ReproBLAS's `double_binned` variable for every MPI process, then we use `binnedBLAS_dbdsum` to collect those into the `local_sum`. After that, we use reproBLAS's method to call an `MPI_Allreduce` with the `binnedMPI_DBDBADD` operator. Finally, we convert the result back to a regular double precision variable and return it.

### 4.4. Reproducible Codegeneration with OP2

Using OP2's source-to-source translator, a user can easily generate reproducible code from an app that already has an implementation using OP2. A few flags are responsible for controlling the mechanisms that allow reproducible code to be generated. In the translator scripts these are: `reproducible`—needed for all methods, `repr_temp_array`—for using temporary arrays, `repr_coloring`—for using reproducible coloring method and `trivial_coloring` which will produce the trivial coloring version. To enable the greedy coloring technique, the `-op_repro_greedy_coloring` command line flag must be used with the application.

### 5. Performance Results

We measured our techniques with four test applications, introduced in Section 3.8. All results are the average of 10 measurements. Table 1 summarizes the details of the different machine setups we used for our measurements.

**Table 1.** Details of the different machine setups.

| Name | CPU | GPU | #Processes per Node | Compiler | OS |
|---|---|---|---|---|---|
| Cirrus-CPU | Intel Xeon E5-2695 (Broadwell) @ 2.1 GHz | n.a. | 18 cores, 2 threads per core per node | icc (ICC) 19.0.0.117 | Red Hat Enterprise Linux 8.1 (Ootpa) |
| Cirrus-GPU | 2.5 GHz, Intel Xeon Gold 6248 | NVIDIA Tesla V100-SXM2-16GN | 20 cores, 2 threads per core, 4 GPUs per node | nvc++ 21.9-0 | Red Hat Enterprise Linux 8.1 (Ootpa) |

All of our methods provide full reproducibility at the expense of additional computations, suboptimal scheduling, or redundant memory usage. The overall cost of these techniques is visualized in Figures 7 and 8 and in Table 2. We compare each run with its original, non-reproducible version. On CPU systems, slowdowns are between 1 and 3.21

times. The difference between the greedy and distributed coloring methods comes down to data reuse and cache line utilization, because of the different number of colors used. The main reason for that is that the data for neighboring elements are located close in memory, but when using coloring, adjacent elements will have different colors, leading to poor utilization. A few examples of the number of colors used are shown in Table 3. While the greedy scheme leads to near-optimal color counts, the parallel scheme yields much higher color counts particularly in 3D. The performance of the trivial coloring scheme is close to the reference, since it uses a similar order of execution to the nonreproducible version, with the only differences around the borders of MPI partitions. Since with the trivial scheme we still require sequential execution within a process, we cannot use additional parallelization techniques, such as CUDA or OpenMP. In contrast, the slowdown on GPUs is more significant, because they are even more sensitive to data access patterns and cache locality than CPUs. In particular, with the usage of the temporary arrays, we have to iterate through the increment data twice, once when populating it and once when gathering the results, each time with a different access pattern. If we optimize for one stage, then the other will suffer from the non-coalesced data accesses. This is even true for the coloring methods. If we reorganize the data in a set according to one map, then later, using another map to the same set, we again obtain inefficient access patterns.

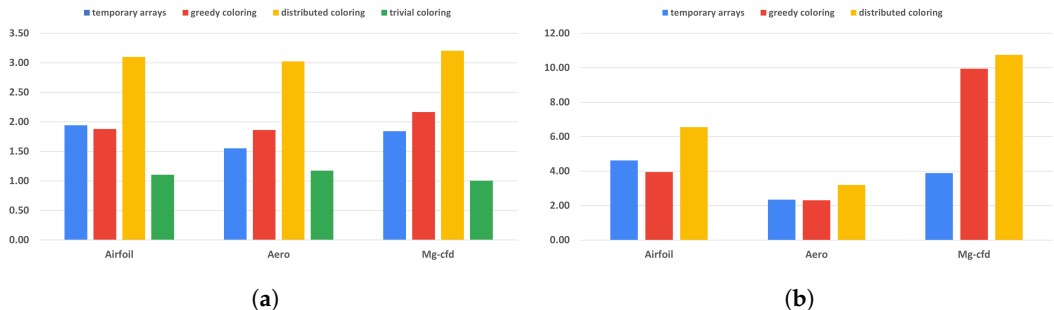

(**a**)　　　　　　　　　　　　　　　　　(**b**)

**Figure 7.** Slowdown effect of the different methods compared to the non reproducible version. (**a**) Using 40 MPI-only processes on the Cirrus machine; (**b**) Using one MPI+CUDA GPU process on the Cirrus-GPU machine.

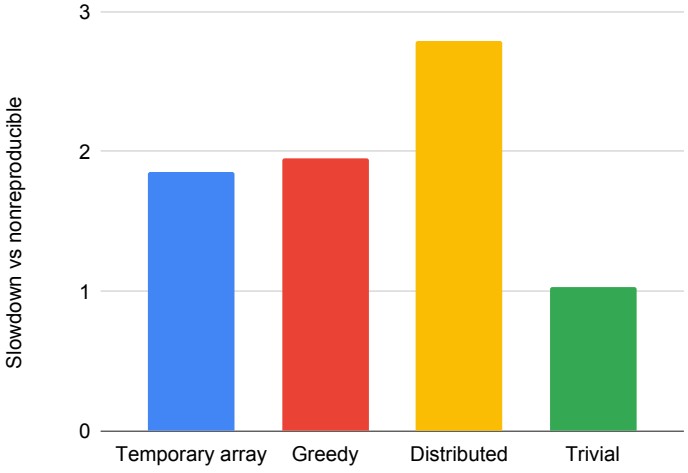

**Figure 8.** Slowdown of Hydra measured on an 8M mesh, 20 iterations, using the Cirrus-CPU machine.

**Table 2.** Memory usage of the reference run and with using the proposed methods in GB.

| App | Non Reproducible | Temporary Arrays | Coloring Method |
|---|---|---|---|
| Airfoil | 0.92 | 1.6 | 1.3 |
| Aero | 2.6 | 3.4 | 2.8 |
| MG-CFD | 7.5 | 14.4 | 9 |

**Table 3.** Number of colors with the different methods on the applications main map.

| App (Map) | Greedy | Distributed |
|---|---|---|
| Airfoil (pecell1) | 4 | 14 |
| Aero (pcell1) | 5 | 17 |
| MG-CFD (edge→node0) | 7 | 19 |

The runtime overhead of the preprocessing preparations of the temporary array and coloring methods against the number of MPI processes are detailed in Figure 9 and using only one process in Table 4.

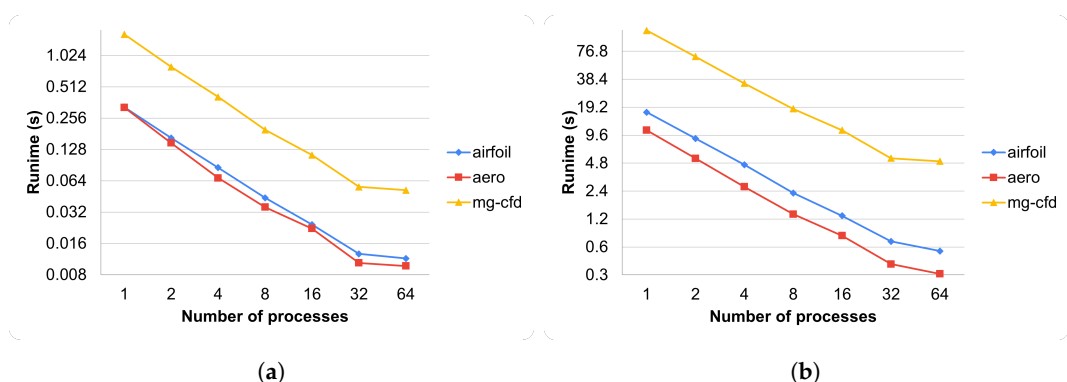

(**a**)  (**b**)

**Figure 9.** Scaling of preprocessing overhead. (**a**) reversed map and temporary array creation time for the temporary array method; (**b**) reversed map creation and distributed coloring time.

**Table 4.** Reversed map creation and greedy coloring time.

| App | Runtime |
|---|---|
| Airfoil | 4.65 s |
| Aero | 1.79 s |
| MG-CFD | 128.88 s |

Figure 10 shows how well the test applications scale with the different methods using one, two, four, and eight nodes on the Cirrus cluster. On the CPU side, all methods have the same parallel efficiency on each application, except the distributed and greedy coloring methods on Airfoil, where we can observe superlinear scaling (Figure 10a) since much of the data used can fit into the cache if they are divided between at least four nodes. We cannot observe this on the temporary array method, because it uses extra memory to store increments separately. Apart from the reductions (discussed in detail below), MPI communications and communication times do not differ between reproducible and non-reproducible. For non-reproducible execution, the communication overhead (as a fraction of total runtime) will become higher using multiple nodes. In the case of reproducible execution, because we spend more time in the colored execution, we spend a smaller fraction of the total time in communications. Therefore, the relative difference is decreasing and the slowdown effect with any method compared to the non-reproducible is less when more nodes are used.

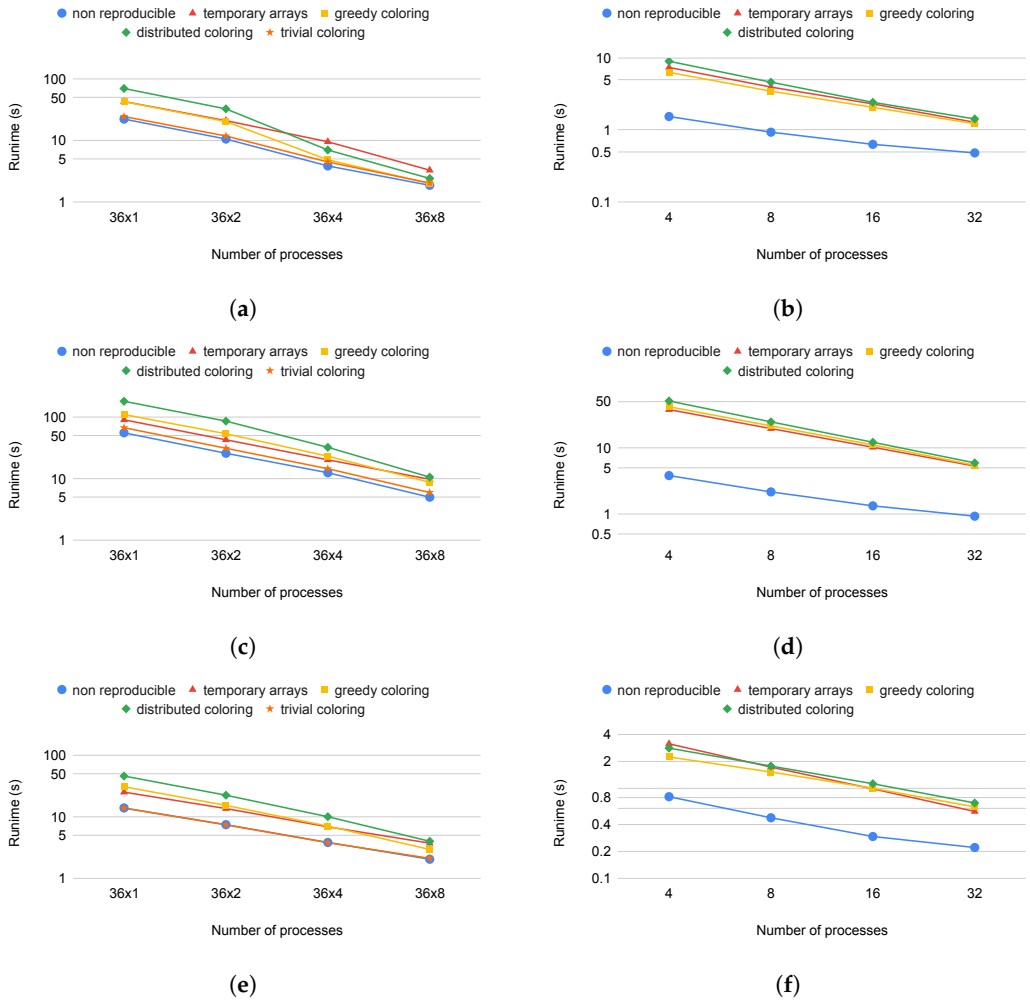

**Figure 10.** Strong scaling measurement of the different methods, using 1,2,4,8 nodes; (**a**) Airfoil, using 36 MPI Intel Xeon CPU processes per node; (**b**) Airfoil, using four Nvidia V100 GPU processes per node; (**c**) Aero, using 36 MPI Intel Xeon CPU processes per node; (**d**) Aero, using four Nvidia V100 GPU processes per node; (**e**) Mg-cfd, using 36 MPI Intel Xeon CPU processes per node; (**f**) Mg-cfd, using four Nvidia V100 GPU processes per node.

We can observe the strong scaling of a reduction kernel in Figure 11. Since all reproducible methods use the same reduction technique, there is no separate measurement for them. Again on CPUs, we can see the superlinear effect as the application fits more and more into the cache. We can also observe that there is an additional cost of the reduction caused by the reproBlas functions. The most significant factor in the cost of reproducible reduction is that we must write all the values to be reduced into a separate array and perform a reduction on it within a process. This leads to extra memory movement compared to the reference version. This is particularly expensive on GPUs because this array must be copied to the host to perform the local summation. `MPI_reduce` is not significantly more expensive.

Using only MPI parallelization, the overhead is quite small (between 1 and 1.12 times). Using shared memory parallelism, it is a bit greater due to the bad cache locality. In some extreme cases, we can even lose the speedup gain from GPUs, our reproducible methods work better on CPU-only systems.

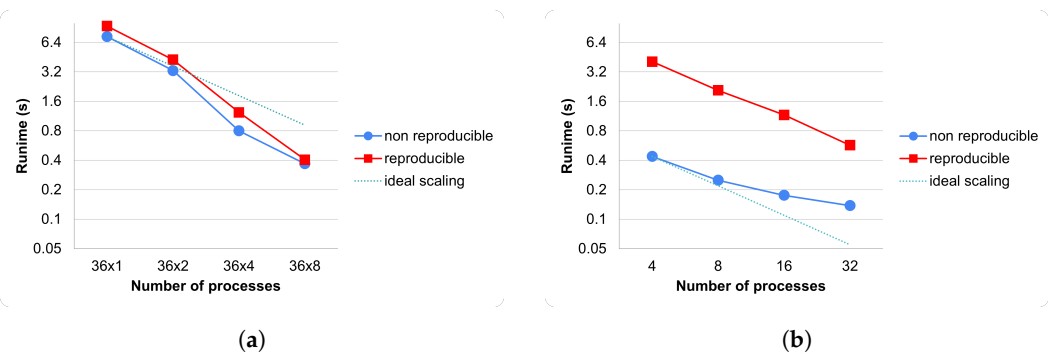

**(a)**                **(b)**

**Figure 11.** Strong scaling measurement of a reduction kernel; (**a**) Airfoil_update on the Cirrus-CPU machine; (**b**) Airfoil_update on the Cirrus-GPU machine.

## 6. Conclusions

In this paper, we examined the non-reproducibility phenomenon that occurs due to the non-associative property of the floating-point number representation on applications defined on unstructured meshes. We compared the differences in results without reproducibility across a range of applications, including Rolls-Royce's production application, Hydra. Non-reproducibility is a widely studied problem; however, we have not yet found an effective solution for distributed systems that could also be applied to arbitrarily partitioned meshes. In this work, we developed a collection of parallel and distributed algorithms to create a plan and then execute it, guaranteeing the reproducibility of the results. Of these, we highlight a graph coloring scheme that gives the same colors regardless of how many parts the graph was partitioned into. We implemented all of our methods in the OP2 DSL and then we showed how they can be automatically applied without user intervention to any application that is already using OP2. We demonstrated that on CPU systems, our methods can achieve bitwise reproducible results with a slowdown between 1 and 3.21 times in various applications, and on GPU systems with a slowdown between 2.31 and 10.7 times due to the modified data access patterns.

While there are alternative methods addressing the issue of reproducible reduction, their complexity is akin to ours and from the perspective of OP2, the choice of method is non-critical. This is why we do not draw comparisons on this aspect, as the time spent on reductions is relatively short. Our work stands out in the development of a generalized method ensuring reproducible execution, applicable to various applications. This is in contrast to other solutions that are application-specific. There are several general methods available. Kahan's method, although popular, does not guarantee reproducibility, just higher accuracy. The most straightforward method involves sorting the elements before adding them. The most general method, perhaps, is the binned method, like in the ReproBLAS library. However, all these methods are more complex and mainly more expensive in computing and/or in memory usage. By leveraging the properties of the unstructured mesh, we can keep the costs low, thus presenting a more efficient solution.

**Supplementary Materials:** All codes developed for this paper can be found at: https://github.com/OP-DSL/OP2-Common/tree/feature/mpi_reproducible_increments-rebase.

**Author Contributions:** Conceptualization, B.S., G.R.M. and I.Z.R.; methodology, B.S., G.R.M. and I.Z.R.; software, B.S.; validation, B.S.; writing—original draft preparation, B.S. and I.Z.R.; writing—review and editing, B.S., G.R.M. and I.Z.R.; supervision, B.S. and I.Z.R.; funding acquisition, G.R.M. and I.Z.R. All authors have read and agreed to the published version of the manuscript.

**Funding:** This research was supported by National Research, Development and Innovation Fund of Hungary (FK 145931), under the FK 23 funding scheme; by Rolls-Royce plc and by the UK Engineering and Physical Sciences Research Council (EPSRC): (EP/S005072/1— Strategic Partnership in Computational Science for Advanced Simulation and Modelling of Engineering Systems—ASiMoV). We thank Leigh Lapworth from Rolls-Royce plc for their valuable feedback.

**Institutional Review Board Statement:** Not applicable.

**Informed Consent Statement:** Not applicable.

**Data Availability Statement:** Data are contained within the article.

**Conflicts of Interest:** The authors declare no conflicts of interest. The sponsors had no role in the design of the study, in the collection, analysis, or interpretation of data, in the writing of the manuscript, or in the decision to publish the results.

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
