# Peer review of "Enabling Bitwise Reproducibility for the Unstructured Computational Motifâ€"

_applsci, doi:10.3390/app14020639_

Round 1
Reviewer 1 Report
Comments and Suggestions for Authors
The topic of the paper is the numerical irreproducibility in the computational motif of unstructured mesh, which occurs due to the non-associative property of the representation of floating-point numbers. The authors proposed several parallel and distributed algorithms to deal with this nondeterminism. The main benefit of the design is the new method of coloring the graph "Distributed reproducible coloring method", which provides an identical coloring result regardless of how many parts the graph is divided into.
Although the article is clearly written and has a good structure, I have comments/questions for the authors that could improve the article if added.
- The core of algorithm No. 3 is lines 14 and 19 (calculate hash value of e in iteration i). In this context, it would be appropriate to mention the hash function.
- In the Conclusions section, it would be appropriate to compare the newly proposed method with some similar methods that are presented in the "Related works" section and discuss the advantages and disadvantages against the solution proposed by the authors.
- In my opinion, it would be more logical to include the "Related works" section right after the "Introduction" section.
Author Response
Dear Reviewer,
Thank you for your insightful comments and suggestions. We appreciate your time and effort in reviewing our paper. We have considered your feedback and made the following changes to improve the quality of our paper:
1. Hash Function in Algorithm No. 3: We have added a mention of the hash function in the context of lines 14 and 19 and cited Robert Jenkins' 32 bit integer hash function.
2. Thank you for your comment. We have revised the conclusion to highlight the comparison of our method with others. While there are alternative methods for reproducible reduction, their complexity is similar to ours and the time spent on reductions is minimal, making the choice of method non-critical for OP2. Our work uniquely provides a generalized method for reproducible execution, unlike other solutions that are application-specific. Despite the existence of several general methods, they are often more complex and expensive in compute and/or memory usage. Our method leverages the properties of the unstructured mesh, keeping costs low and presenting a more efficient solution.
3. Position of the "Related Works" Section: Based on your suggestion, we have moved the "Related works" section to follow the "Introduction" section. We agree that this arrangement is more logical and helps the flow of the paper.
We hope these changes address your concerns. If there are any further points that need clarification, please do not hesitate to ask. We are committed to improving our work and appreciate your valuable input.
Best regards,
Bálint Siklósi
Reviewer 2 Report
Comments and Suggestions for Authors
The article analyzes aspects of non-reproducibility in the application of parallel algorithms related to the construction of meshes, which comprises a wide class of methods linked to the solution of partial differential equations, and presents a special class of methods that avoid these problems for some specific applications. The authors also implement the solutions using a domain specific language and show that these methods are applicable in a generic and automated way.
The main issue of non-reproducibility in the application of parallelized numerical methods is very important, and the literature is quite scarce on this subject. It is important to note that the problem is quite general, as other pde and spde solution (and even related applications in machine learning) methods have the same limitations. So, the motivation of the article is very relevant.
In general, the work is well written, and I believe that the audience is quite broad for the work. The discussions and analyzes are detailed, and the results are solid. Another important point is the implementation provided by the authors, which will certainly be used in other works. I have no additional suggestions for the text or application of the article, and I believe that the work should be approved in its present form.
I think it will be a highly impactful article in this area, and I learned a lot from reading the work.
Author Response
Dear Reviewer,
We greatly appreciate your positive feedback on our paper. Your recognition of the importance of our work in the field of non-reproducibility in parallelized numerical methods is highly encouraging.
We are pleased that you found our discussions detailed, our results solid, and our implementation potentially useful for other works. Your belief in the potential impact of our paper and your kind words about learning from our work are truly motivating.
Thank you for your time and effort in reviewing our paper. Your feedback is invaluable to us.
Best regards,
Bálint Siklósi